# Evaluation of the Hydrophilic/Hydrophobic Balance of 13X Zeolite by Adsorption of Water, Methanol, and Cyclohexane as Pure Vapors or as Mixtures

**DOI:** 10.3390/nano14020213

**Published:** 2024-01-18

**Authors:** Meryem Saidi, François Bihl, Olinda Gimello, Benoit Louis, Anne-Cécile Roger, Philippe Trens, Fabrice Salles

**Affiliations:** 1ICGM, Université de Montpellier, CNRS, ENSCM, 34090 Montpellier, France; meryem.saidi@enscm.fr (M.S.); olinda.gimello@enscm.fr (O.G.); philippe.trens@enscm.fr (P.T.); 2ICPEES, UMR 7515, Université de Strasbourg, CNRS, 25 Rue Becquerel, 67087 Strasbourg, France; francois.bihl@etu.unistra.fr (F.B.); blouis@unistra.fr (B.L.); annececile.roger@unistra.fr (A.-C.R.)

**Keywords:** adsorption, selectivity, zeolite, water, methanol, simulation, co-adsorption

## Abstract

Adsorption isotherms of pure vapors and vapor mixtures of water, methanol, and cyclohexane were studied using a synthesized 13X zeolite (FAU topology), by means of a DVS gravimetric vapor analyzer. These results were validated by GCMC calculations. The surface chemistry of the adsorbent was characterized by the thermodesorption of ammonia, and its textural properties were studied using nitrogen physisorption. The 13X zeolite was found to be strongly acidic (BrØnsted acid sites, Si/Al = 1.3) and its specific surface area around 1100 m^2^·g^−1^. Water was found to be able to diffuse within both the supercages and the sodalite cavities of the FAU structure, whereas methanol and cyclohexane were confined in the supercages only. The water/methanol sorption selectivity of the 13X zeolite was demonstrated by co-adsorption measurements. The composition of the water/methanol adsorbed phase could be calculated by assuming IAST hypotheses. This model failed in the case of the water/cyclohexane co-adsorption system, which is in line with the non-miscibility of the components in the adsorbed state. The sorption isotherms could be successfully simulated, confirming the robustness of the forcefields used. The 13X zeolite confirmed its a priori expected hydrophilic nature, which is useful for the selective adsorption of water in a methanol–water vapor mixture.

## 1. Introduction

In the realm of gas separation, purification, or heterogeneous catalysis, the omnipresence of water, whether as a solvent or in trace amounts, poses a perpetual challenge. Managing the impact of water on these processes necessitates a deep understanding of its interactions with materials, the utilization of advanced characterization techniques to monitor its presence, and the development of innovative material and process design strategies to enhance efficiency while minimizing the undesirable effects of water. Ongoing research in this field aims to discover novel solutions to overcome the enduring challenges associated with the omnipresence of water [1,2,3].

Indeed, material selection plays a pivotal role in these domains, as it is crucial for enabling efficient applications [4]. For instance, addressing climate change requires innovative approaches, and numerous studies have demonstrated the potential for capturing CO_2_ or VOCs in porous materials [2,5,6,7,8,9].

The primary challenge is closely tied to the presence of water vapor within the gas stream, hindering the effective adsorption of CO_2_ or VOCs. The competition between water species, characterized by their polarity, and organic vapors for active sites often renders the process inefficient, with water being deemed a potential “poison” for adsorbents [10]. In certain applications, such as portable devices, saturated adsorbents are discarded, while in static applications like industrial plants, they are regularly regenerated [11]. To address this issue in industrial processes, conventional practice involves the pre-removal of water from hydrocarbon streams using alumina columns, introducing an additional step and escalating costs. Consequently, the quest for alternative adsorbent materials capable of adsorbing VOCs while remaining impervious to moisture becomes crucial, promising to simplify separation processes, enhance the efficiency of adsorbent materials in humid conditions, and significantly reduce overall costs.

Conversely, there are instances where water can augment the capture of certain species by porous materials. Examples include demonstrated effects on xylene separation on Ba^2+^ exchanged faujasite [12], CO_2_ capture in Al-MIL-53 [13], or ammonia adsorption by activated carbons [14]. Consequently, the hydrophilic–hydrophobic balance of the adsorbent emerges as a pivotal factor determining the efficacy of separation/adsorption in humid conditions, in a wide range of concentrations [15]. The interplay between the surface chemistry and confinement effects in porous solids introduces an added layer of complexity, making the hydrophilic–hydrophobic characterization more intricate compared to simple surface evaluations. While literature on this subject remains relatively limited [16,17], existing studies convincingly underscore the significance of the hydrophilic/lipophilic balance in hydrocarbon separation [18].

To delve into this matter, a prerequisite involves a comprehensive understanding of the surface chemistry properties of adsorbents. Among the diverse approaches available, studying the adsorption of molecules with increasing polarity emerges as a relevant method to evaluate the surface chemistry of selected porous materials.

This parameter can be studied both experimentally by vapor sorption [19] and theoretically by GCMC [17]. Among the families of porous materials studied, those that exhibit interesting properties for CO_2_ adsorption include activated carbons [20,21,22,23], zeolites [24], and MOFs [25]. Among these materials, 13X FAU zeolite remains a promising benchmark material [26]. Indeed, this material exhibits cages of different sizes, some of which are only accessible to very small molecules like water [27] (Figure 1).

It is therefore already sensible to adsorb humid flue gases in the 13X zeolite as only water can fit in the sodalite units, thus drying the flue gas, regardless of the polarity of the sorbates. However, it is also important to focus on the desorption of water molecules from this zeolite. Indeed, the regeneration of the zeolite is a crucial aspect to consider since water separation is usually performed in sequences of fixed-bed sorbent reactors with intermediate recycling stages.

The conventional choice of zeolites as sorbents is primarily motivated by their hydrothermal stability and sorption capacity, but their mass transfer limitations due to their microporous systems represent a significant impediment. In this study, we will investigate the sorption of water, cyclohexane, and methanol, as well as perform measurements of co-adsorption of these three molecules on the zeolite 13X both experimentally and theoretically in order to better assess the hydrophilic–hydrophobic balance of the 13X zeolite.

## 2. Materials and Methods

Systems: The synthesis of the 13X zeolite was inspired by the classical route according to the International Zeolite Association recipe found in the Verified Synthesis of Zeolitic Materials [28]. First, 8.30 g of deionized water and 8.30 g of NaOH were mixed until a clear solution was obtained. Then, 8.1 g of Al(OH)_3_ was added to the solution. The latter solution was stirred and heated at 373 K for 15 min with a reflux heating system. Once the milky solution returned to room temperature, 16.9 g of deionized water was added. Moreover, two other solutions containing 50.7 g of deionized water and 4.9 g of NaOH were stirred until a clear solution was obtained. An amount of 18.3 g of Na_2_SiO_3_ was then added, and 8.6 g of the cooled down solution was added to the other. The two last solutions were mixed together inside a polypropylene bottle and stirred for 30 min at room temperature. The polypropylene bottle was sealed and placed in an oven at 363 K for 16 h. Finally, the zeolite was filtered and washed with deionized water until the pH was below 10, then it was dried in an oven at 373 K. After synthesis, the sample was calcined at 450 °C under air for 12 h before use. The final material used in this study is therefore in its Na form.

### Methods

Experiments: Sorption experiments under isothermal conditions were carried out using a Dynamic Vapor Sorption (DVS) Resolution instrument provided by Surface Measurements Systems, based in London, UK. This apparatus facilitated the precise determination of sorption isotherms at various temperatures and across a specified range of partial pressures. The DVS apparatus consisted of two measurement pans, namely the reference and sample holders, suspended from the arms of an ultra-sensitive Cahn magnetic compensation microbalance capable of detecting minute changes in sample mass, down to 0.1 µg.

These measurement pans were connected to the microbalance via hanging wires placed within a connected double chamber, located within a temperature-controlled cabinet. A constant flow of dry nitrogen gas (at a rate of 200 mL/min) was introduced, mixed with another nitrogen stream containing vapor. This setup ensured the maintenance of a consistent partial pressure of vapor. Each experimental run initiated under isothermal conditions from 0% partial pressure and incrementally increased to higher relative pressures in 5% steps before reverting to 0% in reverse order. These incremental steps in partial pressures within the nitrogen stream were established to determine the complete sorption isotherm. The typical duration of a sorption cycle spanned 6 days, with the equilibrium criterion determined based on the slope of the relative mass variation versus time curve. This parameter, represented as dm/dt (with mass in mg and time in minutes), was defined as 0.004%.min^−1^ within a 10 min time frame.

Exemplary graphs illustrating mass uptake versus time can be found in the Appendix A, including data showcasing the slope of mass uptake over time. All experiments were conducted at a constant temperature of 25 °C. The saturation pressures for the mixed solvents were calculated under the assumption that methanol/water mixtures behave ideally across all molar fractions explored. In this context, the total pressure of the mixture can be estimated following Dalton’s law:(1)$$x_{\mathrm{methanol}=\frac{p_{\mathrm{methanol}}}{p_{\mathrm{methanol}}+p_{\mathrm{water}}}}$$

According to the Raoult law, p_methanol_ = x_methanol_ p*_methanol_, in which p*_methanol_ refers to the vapor pressure of pure methanol at the chosen temperature, whereas p_methanol_ refers to the vapor pressure of methanol in the water/methanol mixture. In a symmetric fashion, the same equation can be written for the water/cyclohexane system.

Assuming that methanol and water vapors are perfect gases, the vapor pressure of mixtures can therefore be estimated as:(2)$$p_{\mathrm{mix}}=p_{\mathrm{methanol}}+p_{\mathrm{water}}$$

At saturation, the saturation pressure of the mixture can therefore be written:(3)$$p_{\mathrm{mix}}^{\mathrm{sat}}=p_{\mathrm{methanol}}^{\mathrm{sat}}+p_{\mathrm{water}}^{\mathrm{sat}}$$
where $p_{\mathrm{methanol}}^{\mathrm{sat}}$ is the saturation pressure of methanol in the water/methanol mixture and $p_{\mathrm{water}}^{\mathrm{sat}}$ is the saturation pressure of water in the same mixture. It can be concluded that:(4)$$p_{\mathrm{mix}}^{\mathrm{sat}}=p_{\mathrm{methanol}}^{*,\;\mathrm{sat}}+x_{\mathrm{water}}(p_{\mathrm{water}}^{*,\;\mathrm{sat}}-p_{\mathrm{methanol}}^{*,\;\mathrm{sat}})$$

The relative pressure of the mixture can therefore be written as:(5)$$\frac{p_{\mathrm{mix}}}{p_{\mathrm{mix}}^{\mathrm{sat}}}=\frac{{x_{\mathrm{methanol}}\;p}_{\mathrm{methanol}}^{*}+x_{\mathrm{water}}{\;p}_{\mathrm{water}}^{*}}{p_{\mathrm{methanol}}^{*,\mathrm{sat}}+x_{\mathrm{water}}\;(p_{\mathrm{water}}^{*,\;\mathrm{sat}}\;-\;p_{\mathrm{methanol}}^{*,\;\mathrm{sat}})}$$

The same calculations can be applied for the cyclohexane/water system even though these molecules are poorly miscible in the liquid phase.

The saturation pressures of the pure sorbates were measured at 25 °C in a volumetric sorption apparatus described elsewhere [29]. They were found to be 3200 Pa, 13,059 Pa, and 17,053 Pa for water, cyclohexane, and methanol, respectively.

NH_3_-temperature programmed desorption (NH_3_-TPD) experiments were conducted using an Autochem II apparatus (Micromeritics) equipped with a thermal conductivity detector (TCD). The fresh sample was cleaned at 250 °C for 20 min in He (20 °C/min, 10 cm^3^/min), followed by refreshing at 50 °C. Next, a gas flow containing 5% NH_3_ in He was applied for 30 min (10 cm^3^/min). The lines were purged with He, and then desorption was performed up to 700 °C (20 °C/min, 10 cm^3^/min) and held for 10 min. The powder X-ray diffractogram was obtained using a Bruker D8 Advance, using λ_Cukα_ = 0.15406 nm. Nitrogen adsorption at 77 K was performed using an ASAP 2020 sorption apparatus from Micromeritics after a degassing stage of 12 h at 250 °C under a secondary vacuum (<1.5 mPa).

Simulations: Monte Carlo simulations were performed using homemade software in order to determine the adsorption isotherms for water, methanol, and cyclohexane in the 13X-zeolite structure saturated with Na+ cations. The 13X framework structure adopted in the present study belongs to the space group Fd3m with a unit cell parameter of 24.85 ± 0.03 Å at ambient temperature [30]. The unit cell was composed of 16 hexagonal prisms, 8 sodalite cages, and 8 supercages corresponding to a general chemical formula of M_x/n_Al_x_Si_192−x_O_384_·yH_2_O, where M^n+^ was the compensating cation, x varied between 0 and 96, y represented the number of water molecules in the unit cell, and the atomic Si/Al ratio was between 1.0 and 1.5. The computation was performed for a rigid framework (NVT ensemble) by considering 88 substitutions of Si^4+^ by Al^3+^ distributed randomly (but in accordance with the Loewenstein law) and by incorporating 88 Na^+^ extra-framework cations to approach the composition of the real samples and to reproduce the impact of the cations present in the largest amount within the pores. Calculations were performed using 10^7^ Monte Carlo steps following 2 × 10^7^ equilibration steps. The partial atomic charges used in these calculations were the following Na^+^: 1; Si: 2.4; Al: 1.4; O: −1.2. They were obtained from DFT (density functional theory) calculations to extract Mulliken charges [31] while the 12-6 Lennard-Jones (LJ) atomic parameters were inferred from UFF for the framework and the extra-framework cations [32]. For the adsorbates: the water molecule was considered as rigid by using the four-site charged LJ TIP4P/2005 model [33] while methanol and cyclohexane molecules were also considered as rigid using the united atom (for alkyl chains) models developed in the TraPPE force field for methanol [34] and cyclohexane [35], respectively. The parameters of the adsorbate/adsorbent LJ interatomic potential were then calculated using the Lorentz–Berthelot combining rule. A cut-off radius of 12 Å was applied to all LJ interactions, and the long-range electrostatic interactions were handled by applying the summation Ewald technique.

## 3. Results and Discussion

### 3.1. Material Characterization

As-synthesized 13X zeolite was first characterized by nitrogen adsorption and powder X-ray diffraction. The corresponding results are shown in Figure 2. The nitrogen adsorption–desorption isotherm is typical of type I, according to the IUPAC, thus confirming the microporous nature of the zeolite. From 0.1 p/p°, a very flat plateau can be observed which is the indication of a very limited external surface (88 m^2^·g^−1^, as obtained using the t-plot method), compared to the overall specific surface area. A slight increase at high relative pressure is reminiscent of the occurrence of mesopores, heterogeneous in diameter, in the range of 10 nm in diameter. The thin hysteresis loop is another indication that some large mesopores are present. However, compared to the overall adsorbed amount, these mesopores are only a side textural property, compared to the large extent of microporosity.

As the BET transform of the sorption isotherm is not accurate for microporous materials [36], the Dubinin–Astakhov method was used which has been shown to be more appropriate [37]. The obtained specific surface area was found to be 885 m^2^·g^−^^1^. However, nitrogen is too large to enter the SOD cages and is only able to probe the supercages. This is why the total accessible surface for smaller molecules than nitrogen is likely higher than 885 m^2^·g^−^^1^. This was confirmed by the estimation of the specific surface area by Monte Carlo, reaching a value of 1158 m^2^·g^−^^1^ if all the pores were able to be visited by nitrogen. The micropores volume within the supercages was estimated to be 0.28 cm^3^·g^−^^1^ according to the same method. As the 13X zeolite is crystalline, its porosity is perfectly defined by the arrangement of the SOD cages around the supercages [27]. As mentioned above, the supercages have a diameter of ~1.3 nm with opening windows of ~0.74 nm.

In Figure 2 (right), the typical powder x-ray diffraction pattern of the synthesized 13X zeolite is reported, being in agreement with a reference taken from the Atlas of zeolites types [27]. The comparison between the diffractograms assessed the sole presence of the FAU structure.

In terms of surface chemistry, the thermodesorption of ammonia was performed between room temperature and 700 °C (Figure 3). After deconvolution of the signal trace (black curve), three Gaussian-type curves appear to best fit the original signal (r^2^ = 0.9993). These are relevant to three populations of BrØnsted acid surface sites [38,39]. At low temperature, a large population of weak acid sites (peak desorption at 180 °C) can be seen. At 260 °C, a small population of medium acid sites can be distinguished, whereas at 335 °C a large population of strong acid sites are revealed. In terms of strengths, the repartition is 39% of weak acid surface sites, 18% of medium surface acid sites, and 43% of strong surface acid sites. This is typical of the 13X zeolite acidities, which confirms the usefulness of this material for acid catalysis [40,41]. The comparison with results issued from the literature can allow the attribution of each component: the high-temperature signal can be linked to the interaction of NH_3_ with extra-framework cations, while low- and medium-temperature signals can be related to the interaction with surface OH groups bonded to Si or Al [31].

### 3.2. Pure Vapors Adsorption

The vapor sorption isotherms are depicted in Figure 4 (left). The experimentally obtained curves (shown as squares) demonstrate a high affinity irrespective of the adsorbed vapor type. This was consistent with nitrogen adsorption, despite its lack of specific interaction due to its weak quadrupolar moment. It can be inferred that the interaction governing adsorption in the micropores is high, primarily due to confinement effects. However, the very low relative pressure could not be explored due to the limitations of the DVS machine. The relatively flat plateaus indicate that the external surface is negligible compared to the total surface area of the zeolite.

Using GCMC simulations, the sorption isotherms were calculated and presented in the same figure (as diamonds). The shapes obtained closely match the experimental ones, confirming the zeolite’s high affinity towards the used sorbates. Discrepancies at saturation can be attributed to the fact that the as-synthesized 13X zeolite is not as perfect as the simulation of the crystal or that all pores are not completely accessible to the sorbate molecules in the experiments. To illustrate this point, by simulation, flat saturation plateaus are obtained justifying the saturation of the pores, whereas in the case of the experimental sorption isotherms, some sorption takes place from p/p° = 0.2 onwards, which can be attributed to the presence of large defects in the crystalline structure or mesopores filling at higher pressures. This could already be evidenced upon nitrogen adsorption (see Figure 2 (left)).

The location of the saturation plateaus can also be discussed. At p/p° = 0.6, all the micropores are filled with sorbates. The micropore volumes can therefore be estimated at this relative pressure. In the case of water, the corresponding amount adsorbed (308 mg·g^−^^1^, or 17 mmol·g^−^^1^) is consistent with that found by Kim et al., who found ~15 mmol·g^−^^1^ adsorbed at p/p° = 0.2 [42]. The hydraulic microporous volume can therefore be deduced as 0.308 cm^3^·g^−^^1^, taking the density of water as 0.997 g·cm^−^^3^ at 25 °C. This density value is slightly lower than the theoretical one calculated from the crystal structure and equal to 0.49 cm^3^·g^−^^1^, suggesting that all small micropores are not completely accessible or that the structure possesses defects. However, in the case of methanol, the adsorbed amount at p/p° = 0.6 is 255 cm^3^·g^−^^1^ which corresponds to a hydraulic volume of 0.285 cm^3^·g^−^^1^, taking 0.7786 as the density at 25 °C. As mentioned above, water is able to visit all of the cavities of the zeolite (supercages and SOD cages), whereas methanol and cyclohexane can only sit within the supercages. Indeed, the sodalite windows cages are 2.8 Å in size, whereas the kinetic diameters are 2.60 Å, 3.60 Å, and ~6 Å for water [43], methanol [44], and cyclohexane [45], respectively. It is therefore expected that the mass uptake is higher in the case of water. The difference between these values is therefore mostly located in the sodalite cavities, which therefore represents ~9% of the total adsorbed amount. By MC calculations, the ratio between supercage and sodalite volumes can be estimated as 12%, thus being consistent with the experimental results. Hence, this confirms that solely water molecules are able to visit the SOD cavities.

The theoretical enthalpic profiles of the pure components indicate that water and methanol undergo high interaction at low adsorbed amounts. Indeed, the extrapolated values at θ = 0 reach around −135 kJ·mol^−^^1^ and −115 kJ·mol^−^^1^ for water and methanol, respectively. This is in line with the expected hydrophilicity of the 13X zeolite. The higher value obtained for water can be understood by considering that its confinement in the sodalite cavities generates a stronger interaction compared to those in the supercages. This questions the regenerability of the 13X zeolite, as a high temperature should be used for recovering the whole porosity of the material after water vapor sorption. Upon higher adsorbed amounts of water or methanol, the enthalpy decreases which indicates that these species interact with less reactive sites, likely far from the walls of the zeolite. The enthalpic profile obtained with cyclohexane is really different in several ways. Indeed, the sorption enthalpy at zero coverage can be extrapolated to −60 kJ·mol^−^^1^, which is very low compared to that of the polar probes. Furthermore, this value slightly increases up to −80 kJ·mol^−^^1^, before decreasing to its enthalpy of liquefaction. This unusual result can be interpreted by considering that cyclohexane is adsorbed in a cooperative fashion in the supercages, i.e., the very first molecules interacting with the next ones on the surface of the supercages. Once the microporosity is filled with cyclohexane, sorption takes place on the external surface of the 13X zeolite, resulting in the liquefaction of cyclohexane at saturation (−30.5 kJ·mol^−^^1^ at 25 °C). A complementary picture can be obtained by discussing simulated adsorbed phases for the three systems, as shown in Figure 5.

The interpretation of the snapshots (see Figure 5) corresponding to the most plausible configurations with the lowest energy shows that the polar molecules strongly interact with both the extra-framework cations and the framework confirming the hydrophilic character of the zeolite. Indeed, for water, the distances of interaction between guest molecules and the framework or the cations were found to be equal to 1.6–1.8 Å and 2.2–2.5 Å, respectively, whilst the interaction between water molecules leads to distances of around 2–2.2 Å. Regarding methanol, the methanol-framework (resp. methanol–extra-framework cations) interaction distances are 1.8–2.0 Å (resp. 2.5–2.7 Å), while the interaction distances between methanol molecules are close to 2.3–2.5 Å. Again, these results confirmed that the zeolite favors the interaction with polar molecules. In contrast, interaction distances with cyclohexane molecules and zeolite are longer (3.3–3.5 Å for both cyclohexane and extra-framework cations or the zeolite framework) and cyclohexane–cyclohexane distances are longer than 3.5 Å, thus confirming the lower attraction of the framework for non-polar sorbates.

### 3.3. Mixtures Sorption Isotherms

For each system, three compositions were tested in order to precisely determine the hydrophilic/lipophilic balance of the 13X zeolite. The water/methanol mixture’s sorption isotherms are reported in Figure 6 (left). Two key features can be elucidated: (i) the sorption isotherms of the mixture exhibit a shape closely resembling those obtained for individual components, and (ii) the saturation plateaus in the mixture sorption isotherms are positioned between those observed for the pure systems. The initial observation implies that water and methanol are adsorbed in the 13X zeolite, irrespective of its surface sites, particularly the acidic ones. It can be inferred that confinement effects play a pivotal role in governing the sorption behavior of these sorbates. The latter observation implies that water and methanol are somehow adsorbed in proportion to their respective compositions in the vapor phase. 

There is no apparent synergistic effect for which cooperative adsorption could be observed. Indeed, in cases of synergistic adsorption, the adsorbed amount for the mixture vapors would be expected to surpass that observed for the individual components. However, this is not observed in the current study. We endeavored to assess the composition of the adsorbed phase for each mixture sorption isotherm. Indeed, the adsorbed mass is a global measurement of what is actually in the micropores.

For this evaluation, we decided to focus on the supercages filling only, as neither methanol nor cyclohexane can enter the SOD cavities. Such a hypothesis can be reinforced by the density of the presence obtained for simple vapors from Monte Carlo simulations illustrating the localization of guest molecules in the 13X zeolite pores (see Figure 7). Based on the comparison between water and nitrogen, we assumed that water could adsorb 255 cm^3^·g^−1^ at saturation of supercages, instead of 270 cm^3^·g^−1^, which includes the sodalite cavities. We first considered the methanol/water mixture sorption.

In Table 1, we reported the adsorbed masses at p/p° = 0.6 for each sorption isotherm along with the estimation of the masses adsorbed in the sodalite cages and within the supercages.

For these calculations, we assumed that only water could enter the sodalite cavities. As the sodalite cavities represent ~9% of the porous volume (as determined previously according to the experimental results), we assumed that 9% of the adsorbed amount of water would be in the sodalite cavities (i.e., 24.5 mg·g^−1^). We also assumed that the sodalite cavities would always be filled with water in the case of the water/methanol mixtures, regardless of the vapor mixture content. The mass adsorbed in the supercages was therefore given by the difference between the total adsorbed mass at p/p° = 0.6 and the adsorbed mass in the sodalite cavities. The mass adsorbed in the supercages was calculated according to the following Equation (6):(6)$$M_{\mathrm{supercages}}=M_{\mathrm{Water}}^{\mathrm{sat}}\;\text{×}\;{\%}_{\mathrm{Water}}^{\mathrm{adsorbed}}+M_{\mathrm{MeOH}\;}^{\mathrm{sat}}\;\text{×}\;{\%}_{\mathrm{MeOH}}^{\mathrm{adsorbed}}\;$$
where $M_{\mathrm{Water}}^{\mathrm{sat}}\;$ is the mass of pure water adsorbed in the supercages (283.5 mg·g^−1^) at 0.6 p/p°; $M_{\mathrm{MeOH}}^{\mathrm{sat}}$ is the mass of pure methanol adsorbed in the supercages (224.9 mg·g^−1^) at 0.6 p/p°.

The Equation (6) is therefore based on the ideal adsorbed solution theory (IAST), in which the adsorbed phase is considered as an ideal mixture of two components [46].

The percentages of water and methanol were adjusted to ensure that M_supercages_ + M_sodalites_ = M_Total_ which is defined for each composition of the vapor phase. These percentages are therefore the composition of the adsorbed phase in the supercages. This composition can be different from the relative proportion of the two components in the vapor phase in equilibrium with the adsorbed phase.

The results presented in Table 1 can be discussed in terms of selectivity. It can be seen that the proportion of water in the adsorbed phase is higher, compared to that in the vapor phase which is the indication that water is adsorbed preferentially to methanol. This result is consistent with the modelled enthalpy of adsorption of these molecules on the zeolite 13X which could be broadly extrapolated between −135 kJ·mol^−1^ and −115 kJ·mol^−1^ for water and methanol, at zero coverage. These results are consistent with the literature [47,48]. It must be remembered that in the case of water, the enthalpy of adsorption includes the sodalite cavities in which only water can be adsorbed. As their size is very small, confinement effects are likely responsible for the higher sorption enthalpies observed.

For investigating the hydrophilic/lipophilic balance of the 13X zeolite, we also used cyclohexane/water mixtures, even though these components are not miscible in the liquid phase. The sorption isotherms of the pure components and their mixtures are presented in Figure 6 (right). The shape of the sorption isotherms is similar to those obtained with water/methanol mixtures. Indeed, they all belong to type I, indicative of a good affinity between water/cyclohexane mixtures and the 13X zeolite, with saturation, once the micropores are filled with cyclohexane and/or water molecules. However, compared to the trend observed in the case of the water/methanol systems, the relative location of the saturation plateaus is different in the case of the mixtures. We therefore decided to apply the same methodology to try to determine the composition of the adsorbed phase, based on the saturation plateaus of the pure components. In this case, we also decided that water could enter the sodalite cavities, regardless of the presence of cyclohexane in the supercages. The results are reported in Table 2.

In contrast to the trend observed in the case of the water/methanol system, no clear relationship between the composition of the vapor phase and that of the adsorbed phase could be established in the case of the water/cyclohexane system. Notably, when the vapor phase consisted of 66% water, the composition of the adsorbed phase was higher at 71.5%. This suggests a predominance of hydrophilic interactions in the adsorbed phase, thereby lowering the sorption of cyclohexane. Conversely, in the case where the vapor phase comprised 50% water, the adsorbed phase exhibited a cyclohexane composition at 77%. This result is unexpected as the enthalpy of water is greater compared to that of cyclohexane. Furthermore, for a higher proportion of cyclohexane in the vapor phase (66%), the calculation of the adsorbed phase composition proved unattainable. Even under the assumption that the supercages were exclusively filled with cyclohexane (225 mg·g^−1^), while the sodalite cavities were filled with water (24.5 mg·g^−1^), the total mass adsorbed surpassed the expected value of the saturation plateau (233 mg·g^−1^). A plausible interpretation for this observation is that the presence of cyclohexane in the cavities hampers water accessibility to the sodalite cavities, despite their notable interaction with water. This highlights the nature of the adsorbed phase which remains questionable as it is made of non-miscible components in the liquid phase even though they can form a homogeneous vapor phase.

## 4. Conclusions

Adsorption isotherms for pure vapors and vapor mixtures, including water, methanol, and cyclohexane, were investigated using a synthesized 13X zeolite. This exploration employed a DVS gravimetric vapor analyzer alongside various surface chemistry techniques. The obtained results were corroborated through grand canonical Monte Carlo (GCMC) calculations. The adsorbent’s surface chemistry was characterized by the thermodesorption of ammonia, while its textural properties were examined using nitrogen adsorption at 77 K. The 13X zeolite exhibited significant acidity (BrØnsted acid sites, Si/Al = 1.25) with a specific surface area of approximately 1100 m^2^·g⁻^1^.

Water demonstrated the ability to access both the supercages and the sodalite cavities of the 13X zeolite, whereas methanol and cyclohexane were confined solely to the supercages. The water/methanol selectivity of the 13X zeolite was confirmed through co-adsorption measurements. The composition of the water/methanol adsorbed phase was determined using ideal adsorbed solution theory (IAST) assumptions. However, this model proved inadequate for the water/cyclohexane co-adsorption system, aligning with the immiscibility of components in the adsorbed phase. GCMC simulations facilitated the calculation of sorption interactions in the 13X zeolite for different systems. The sorption isotherms were successfully simulated, affirming the robustness of the employed forcefields. The 13X zeolite was confirmed to be a hydrophilic material, not only because of the sodalite cavities, as proved by simulation of the enthalpies of adsorption of the three sorbates at low relative pressure. The regenerability of the zeolite is however questionable, as the enthalpy of adsorption of water is rather high, due to the interactions in the SOD cavities. These results could prove useful for the selective adsorption of water in a methanol–water vapor mixture. The simulation of the vapor mixtures’ sorption isotherms could help to better interpret the experimental results. The corresponding results will be published in the near future.

## Figures and Tables

**Figure 1 nanomaterials-14-00213-f001:**
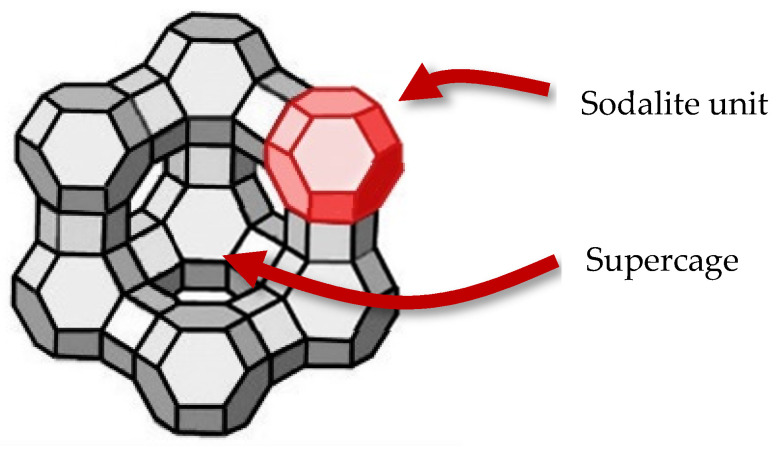
Representation of the 13X zeolite showing the supercage of a 1.3 nm inner diameter and sodalite units of a 0.74 nm inner diameter (as determined from the PSD calculated from the crystal structure, illustrated in Appendix A).

**Figure 2 nanomaterials-14-00213-f002:**
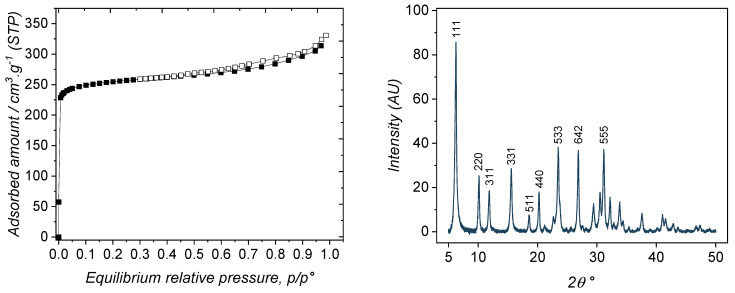
(**left**) Nitrogen sorption isotherm on the 13X zeolite performed at 77K; (**right**) X-ray diffraction pattern of the 13X zeolite.

**Figure 3 nanomaterials-14-00213-f003:**
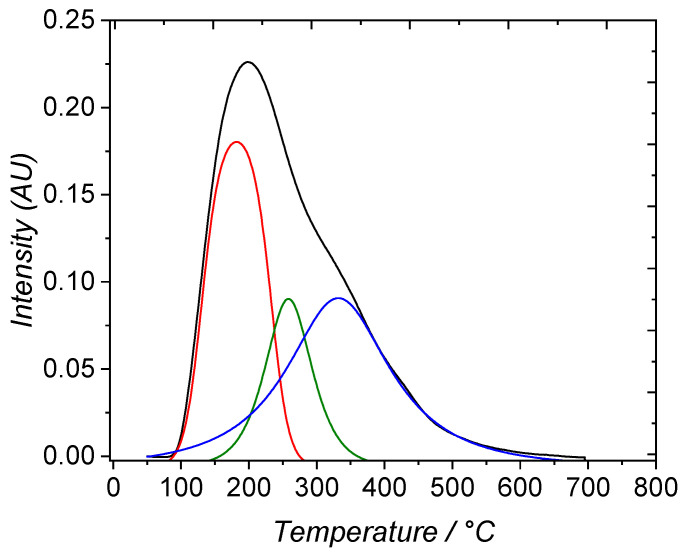
Temperature programmed desorption of NH_3_ on the 13X zeolite. The deconvolution of the original curve is shown in red, green and blue colors for the three acid sites populations.

**Figure 4 nanomaterials-14-00213-f004:**
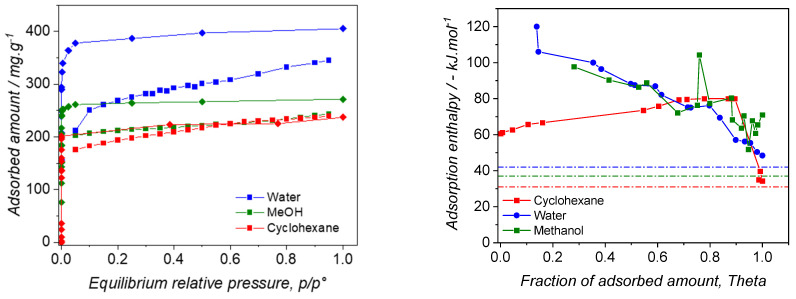
(**left**) Pure vapors adsorption isotherms obtained at 25 °C; (squares) experimental results; (diamonds) simulated results; (**right**) Simulated enthalpy of adsorption of pure water, methanol, and cyclohexane on 13X at 25 °C. Dashed lines refer to the enthalpy of condensation of the pure vapors.

**Figure 5 nanomaterials-14-00213-f005:**
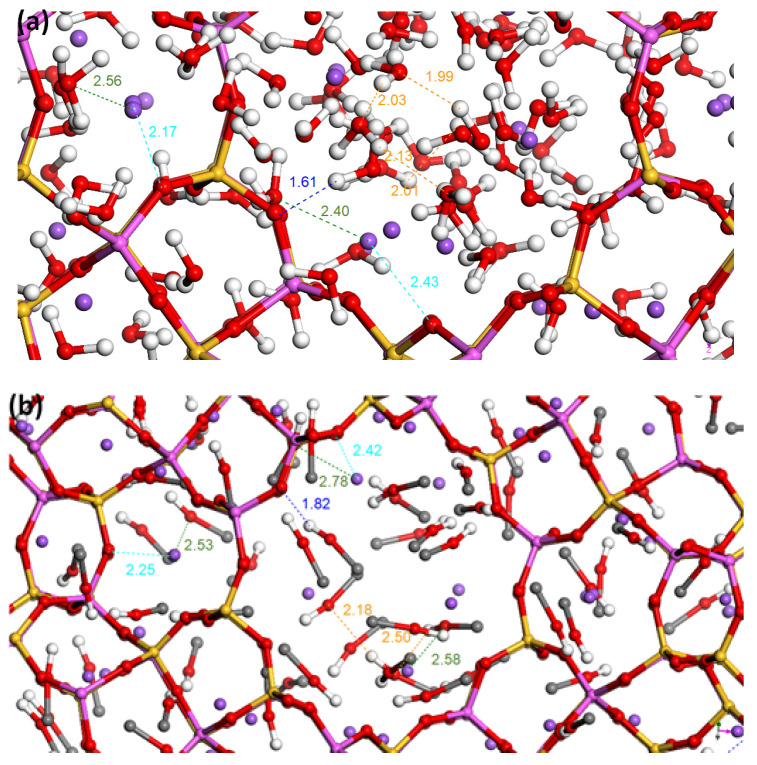
Snapshots obtained at the saturation of the simulated sorption isotherms at 25 °C in the case of (**a**) water sorption; (**b**) methanol sorption; (**c**) cyclohexane sorption. (Purple ball: Na^+^; red ball: O; black ball: C; white ball: H). Full size figures are presented in Appendix A in the supporting information. The blue, green, cyan and orange lines correspond to sorbate-zeolite framework, extra-framework cation-sorbate, extra-framework cation-zeolite framework and sorbate-sorbate interactions respectively.

**Figure 6 nanomaterials-14-00213-f006:**
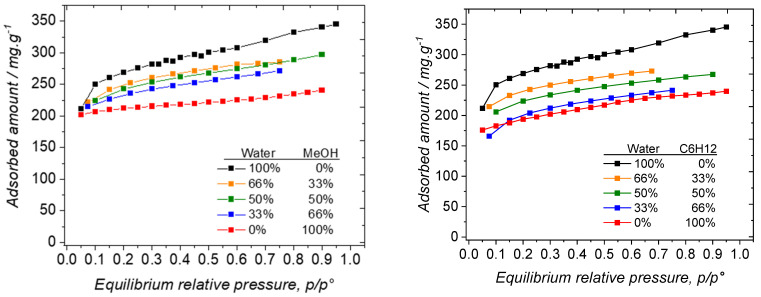
(**left**) Sorption isotherms water/methanol mixtures on 13X at 25 °C; (**right**) Sorption isotherms water/cyclohexane mixtures on 13X at 25 °C. As explained in the manuscript, the relative pressures for the mixtures have been calculated by assuming that the Dalton and the Raoult laws apply.

**Figure 7 nanomaterials-14-00213-f007:**
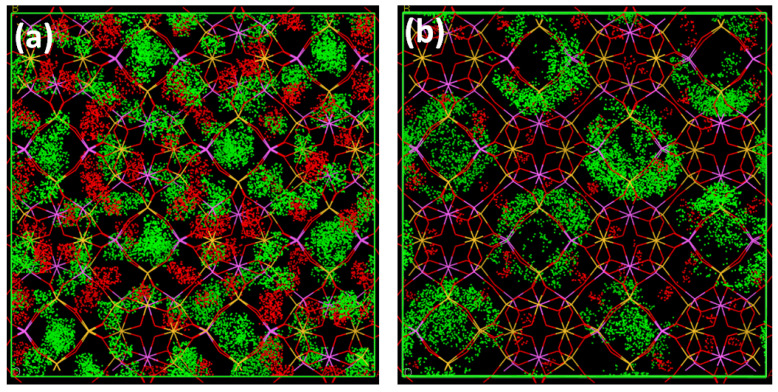
Density of presence for cations (red) and sorbates (green) for water (**a**) and cyclohexane (**b**). Similar results to those obtained with cyclohexane are found for methanol.

**Table 1 nanomaterials-14-00213-t001:** Water/methanol mixture adsorption at p/p° = 0.6. SOD stands for sodalite cages adsorbed amount in mg·g^−1^, Scages stands for supercages adsorbed amounts in mg·g^−1^. Total is the sum between Sod and Scages.

Molar Water Vapor %	Plateau mg/g	SOD mg/g	Scages mg/g	Total mg/g	Molar Adsorbed Water %
100	308.0	24.5	283.5	308.0	100
66	281.6	24.5	256.5	281.0	75.6
50	274.3	24.5	249.9	274.5	64.0
33	261.8	24.5	237.2	261.7	49.5
0	224.9	0	224.9	224.9	0

**Table 2 nanomaterials-14-00213-t002:** Water/cyclohexane mixture adsorption at p/p° = 0.6. SOD stands for sodalite cages adsorbed amount in mg·g^−1^, Scages stands for supercages adsorbed amounts in mg·g^−1^.

Molar Water Vapor %	Plateau/ mg/g	SOD/ mg/g	Scages/ mg/g	Total/ mg/g	Molar Adsorbed Water %
100	308.0	24.5	283.5	308.0	100
66	270.1	24.5	245.5	270.0	71.5
50	253.2	24.5	228.5	253.0	23.0
33	233.0	24.5	225.0	249.5	Undefined
0	225.0	0	224.9	224.9	0

## Data Availability

Data can be made available upon request to the corresponding author.

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
