# Peer review of "Evaluation of the Hydrophilic/Hydrophobic Balance of 13X Zeolite by Adsorption of Water, Methanol, and Cyclohexane as Pure Vapors or as Mixtures"

_nanomaterials, 2024, doi:10.3390/nano14020213_

Round 1

Reviewer 1 Report

Comments and Suggestions for Authors

Adsorption isotherms of pure vapors and vapor mixtures of water, methanol and cyclohexane were studied using a synthesised 13X zeolite by means of a DVS gravimetric vapor analyzer, whose results were comforted by GCMC calculations. The manuscript are suitable for publication on the Journal.

Reviewer 2 Report

Comments and Suggestions for Authors

Saidi and co-workers provide an investigation on the adsorption behavior of zeolite X. They focus on the quantity of adsorptives inside the material and provide co-adoption measurements, which are, to my knowledge, not or only scarcely available. Thus the study is of general interest. All in all, I suggest to accept the study after minor changes noted.

The Introduction’s focus is, what the authors intend to do with these materials in subsequent studies (mainly CO2 reduction). But CO2 is not part of this investigation. Instead, the reader expects information about the adsorption of water, methanol, and cyclohexane on this material. Thus, instead or in addition, I suggest to better discuss the actual adsorption behavior of zeolites for one or multiple adsorptives presented in the introduction.

Especially, since in the following the study does not provide or link to any spectroscopy information about the surface species that form or the form of adsorption, respectively.

For example, the investigated zeolite is acidic, in contrast to other named benchmark materials like typical MOFs or carbon. Thus, it would be good to provide a short summary on the fact that the adsorption of water and methanol changes on this acidic surface. And while co-adsorption measurements are scarce, recent studies on the form of adsorption of water or methanol on surfaces are found.

Under materials and methods, strange sentences like: “Methods. Sorption experiments. Experiments.” are found. I suggest to clarify.

Was the material in H- or Na-form? I found no information about a post-synthesis ion exchange.

Comments on the Quality of English Language

Good.

Reviewer 3 Report

Comments and Suggestions for Authors

To Authors

This manuscript deals with the evaluation of the hydrophilic/hydrophobic balance of the 13X aluminosilicate zeolite (Si/Al=1.3) by adsorption of sole pure water, methanol and cyclohexane vapors, as well as their binary mixtures. The authors have synthesized the material, then confirmed its crystal form (XRD) and studied its textural parameters (low-temperature nitrogen adsorption-desorption). The chemical character of the material was studied by the TPD of ammonia. The main goal of this study was to investigate the adsorption properties of the microporous solid toward adsorption of water and methanol (both polar and mutually miscible adsorptives), as well as nonpolar cyclohexane. The ‘host-guest’ adsorbent-adsorptive systems were studied for single adsorptive as well as binary ones: water-MeOH and water-cyclohexane under the assumption of the Ideal Adsorbed Solution Theory. The results of the real vapor’s isotherms measured using the Dynamic Vapor sorption were confronted against the GCMC simulations (note: the latter one was calculated using a home-built software). Here, the aim was to elucidate the spatial, say, ‘arrangement’ of the adsorptive within the confined nanospaces of the adsorbent (both, the narrow SOD chambers and the broader supercages). It was found that water can enter both the types of the pores, notwithstanding their geometry and dimensions. Meanwhile, both organic adsorptive are not capable to visit the narrow SOD cavities. This is of paramount importance when talking about the work of the adsorbent in the removal of mixed (particularly: humid) adsorptives.

The overall layout of the work is sufficient, the topic may be interesting to the audience in the field, however, the general conclusion that 13X zeolite has hydrophilic nature and, therefore, adsorbs preferably water, is not novel. Nevertheless, the compatibility of the theoretical consideration with the real data is interesting.

The language is professional, and the empirical results as well as the simulations seem to be reasonable. There are, however, certain minor issues to be corrected/clarified/supplemented. They are the following:

1/ Lines 134-135: ‘Exemplary graphs illustrating mass uptake versus time can be found in the supple-134 mentary materials (Fig. S1, Supplementary Material)’, but Fig. S1 shows the ‘Snapshots obtained at the saturation of the simulated sorption isotherms’.

2/ The authors nicely discuss the external surface area of the zeolite under study, but they do not show its value. As it can be easily calculated from the slope of the adsorption isotherm, it would be nice to show it explicitly.

3/ Fig. 2. (right) shows the XRD of the sample, which evidences the crystal arrangement of the material. It would be nice to see the reference XRD as well as the Miller indices of the reflections.

4/ Lines 243-244: ‘As mentioned above, the supercages have a diameter of ~1.3 nm with opening windows of ~0.74 nm.’ Could you provide any empirical proof for this statement?

5/ Fig. 4. (left) what is the reason that the measured and simulated isotherms for cyclohexane are mutually most similar, whilst for water they are so different (in terms of the plateau)?

6/ Line 318: ‘the sodalite windows cages are 2.8 A in size’, but Fig. 1. shows different value.

7/ Line 442: ‘The volume adsorbed in the supercages’, not the mass adsorbed…?

Please, address these issues. With such corrections, I will recommend publication of this article in Nanomaterials.
